# Prevalence of pterygium and its associated factors among adults aged 18 years and above in Gambella town, Southwest Ethiopia, May 2019

**Sofonias Addis Fekadu** *, **Abel Sinshaw Assem**, **Nebiyat Feleke Adimassu**

Department of Optometry, College of Medicine and Health Sciences, University of Gondar, Gondar, Ethiopia

☯ These authors contributed equally to this work.
* addissofi2007@gmail.com

## Abstract

### Introduction

A pterygium is a wing-shaped fibro-vascular growth of conjunctiva on the superficial cornea/conjunctiva. It is an elastotic degeneration of conjunctival stroma mainly due to Ultraviolet light exposure. The prevalence of pterygium varies in different environmental conditions. Its magnitude varies widely from 1.1% to 53% globally and in Ethiopia, it reaches from 8.8% to 38.7%.

### Objective

To determine the prevalence of pterygium and its associated factors among adults aged 18 years and above in Gambella town, Southwest Ethiopia, 2019.

### Methods and materials

A community based cross-sectional study was conducted from April 15 to May 3, 2019, in Gambella town. A total of 402 study participants were selected using a systematic random sampling technique. A pre-tested semi-structured questionnaire, torch, and magnifying loops were used to collect data. The data was entered into epidemiological information 7.1 and exported to statistical package for social science version 20 for analysis. The binary and multivariate logistic regression analysis model was fitted to identify factors associated with pterygium. Odds ratio with respected 95% CI was used to identify the direction and strength of association.

### Results

A total of 400 participants were examined with a response rate of 99.50%. The mean age of the study participants was 39.9±9.8years. The prevalence of pterygium among adults aged 18 years and above in Gambella town was 127(31.80%), (95% CI: 27.3, 36.3). male sex (AOR = 2.10 (95% CI: 1.26, 3.45), sunlight exposure (AOR = 6.86 (95% CI: 4.00, 11.79) and

**Data Availability Statement:** All relevant data are within the manuscript and its Supporting Information files.

**Funding:** The authors received no specific funding for this work.

**Competing interests:** The authors have declared that no competing interests exist.

outdoor works (AOR = 2.10 (95% CI: 1.21, 3.60) were positively associated with pterygium whereas wearing sunglass/hat was a protective factor.

## Conclusion

The prevalence of pterygium was high among adults living in Gambella town. Wearing sunglass/hat and reducing exposure time to the sun was important to reduce the development of pterygium in adults.

## Introduction

A pterygium is a wing-shaped fibro-vascular growth of conjunctiva on the superficial cornea/conjunctiva and is an elastotic degeneration of conjunctival stroma mainly due to Ultraviolet (UV) light exposure [1, 2]. It is one of the most common disorders of the cornea that degrades the visual performance of the eye [3]. The prevalence of pterygium varies from 1.1% to 53% globally in different environmental conditions [4–7] and in Ethiopia, it reaches from 8.8% to 38.7% [8]. It has a higher magnitude in countries that are closer to the equator due to a higher level of outdoor UV exposure [9].

A fibro-vascular growth that encroaches the nasal /temporal cornea causes visual disturbance/reduction due to astigmatism [6, 10]. Pterygium is asymptomatic at the beginning [11] but, as it progresses it decreases visual function secondary to loss of corneal transparency within the pupillary area and increases irregular corneal astigmatism [15]. If a pterygium is left untreated, it is potentially blinding disease that causes personal, social, and economic cost [12–14] and accounts for 2.2% of total blindness causes in at least one eye [15]. Redness, ocular irritation, dryness, and ocular discomfort are common ocular symptoms raised by patients with pterygium [16, 17]. As fibro-vascular growth reaches at the limbus, mid iris, pupil margin, and covers the whole pupil, clinically they are graded as grade 1,2,3, and 4, respectively [17, 18].

Many studies done in different areas of the world indicated that exposure to ultraviolet radiation, geographical latitude near to the equator, outdoor activity [19], aging, male gender [20], and dusty environment are possible risk factors of pterygium [7, 11, 21]. As ultraviolet radiation is the major risk factor to pterygium [2, 9] wearing sunglasses with UV protective lens and brimmed hats are recommended [9, 11]. Refractive correction for astigmatisms, artificial tears, non-steroidal anti-inflammatory eye drops, and surgery [16, 17, 22] are the most common management options of pterygium depending on its stage [9]. The indications for pterygium surgery include; unusual occurrence to the cornea (cosmetic issue), if it extends close to the visual axis, when it restricts eye movement, when it causes visual loss from astigmatism, and when a patient reports growth of the pterygium [16].

Most of the studies in different areas did not show the determinants of the disease at the community level and as the best of our search, there are no studies that determine even the magnitude of pterygium in the study area. Therefore, the main aim of this study is to determine the prevalence of pterygium and its associated factors among adults aged 18 years and above in Gambella town, Southwest Ethiopia.

## Methods and materials

### Study design, setting and population

A community based cross-sectional study was applied from April 15 to May 3, 2019. This study was carried out in Gambella town which is 777 kilometers away from Addis Ababa,

capital city of Ethiopia. The town has dry, hot, and humid environment located in 527 meters above sea level. It has a population size of 76,140 with adult population of 46,787 and has 11,092 households. The study includes all adults aged 18 years and above who lives in the town. Adults with severe illness and sever ocular injuries during data collection were excluded from the study.

## Sample size determination

The sample size was calculated using a single population proportion formula for the proportion of pterygium. By taking a similar study done in Ethiopia with a proportion of 38.7% [8], 95% confidence level, 5% margin of error, and 10% non-response rate the final minimum sample size for this study was determined to be 402.

## Sampling techniques and procedures

The Participants of this study were selected by using systematic random sampling technique. A list of households was taken from each of the 7 kebeles administration (smallest administrative unit). After calculating the interval as "28" by dividing the total households in the town to the sample size calculated (402), lottery method was used to select the starting house. Every 28th house was used to conduct the study by assuming that each house contains at least one adult.

## Operational definitions

**Sun exposure.**  Participants who are exposed 5 hours and above per day were considered exposed whereas those participants who exposed below 5 hours per day were considered as non-exposed [23–25].

**Outdoor workers.**  It includes; farmers, gas/electric workers, miners, recreational workers, watermen, postmen, welders, and others whose activities are away from their house/office [16, 26] for five and more hours per day for five years and above [25].

**Pterygium grading.**  Grading was done according to the position of head of the fibrovascular growth to the cornea.

Grade I: the head of the pterygium touches/reaches the limbus.

Grade II: the head of the pterygium reaches between the limbus and undiluted pupil margin.

Grade III: the head of the pterygium reaches the pupil margin.

Grade IV: the head of the pterygium reaches within the pupil area [23, 26].

**Dust exposure.**  A person having at least a history of one episode of exposure to dusty environment that causes their eye to be irritated, itchy and red within the past 6 months [27].

**Sun glass use.**  Participants who used any type of glass (cosmetics, general and specially treated) for the purpose of protection from sun during their outdoor activities [28, 29].

## Data collection tools and procedure

The data collection tool (questionnaire) was developed by reviewing different works of literature on similar studies [8, 11, 13, 24]. Binocular magnifying loops and ophthalmic torches were used in the examination part of the study to detect the presence of pterygium. The data was collected using two optometrists and one ophthalmic nurse under one supervisor. Appropriate training was given to the data collectors on how to collect data, the ways on

how approach to study participants and examination techniques of pterygium before data collection.

The data collectors introduced themselves to the head of the house when they reach the selected houses. After explaining the purpose of the study and getting a written consent, data were collected from adults by interviewing and physical examination. Double check up was carried out, when the selected houses were closed during the data collection. If the houses did not open in the next days again, next houses were included in the study. Lottery method was applied to select participants in a house that contains more than one adult. After completing the interview each study participants were examined by using binocular magnifying loops and ophthalmic torches. All participants were advised about the characteristics, risk factors and possible management options of pterygium. Finally, study participants who were examined as pterygium were referred to the nearby hospitals. In addition, other suspected eye diseases were also advised and referred accordingly depending on the nature of the cases.

## Data quality assurance

The English version of questionnaire was translated to a similar form of Amharic version and back to English by language professional to increase the accuracy and consistency level of the questionnaire. The Supervisor and data collectors had given appropriate and sufficient training on the ways of face to face interview as well as examination techniques of pterygium in the community with the help of principal investigator. The questionnaire was pre-tested in five percent of the sample (21 participants) in Abol town which is 15 kilometers away from Gambella town before the data collection period. Modifications were done accordingly to ensure the appropriateness and common understanding of the study material/tool. Data quality was controlled by a close follow-up of the data collection process by the supervisor. Completeness of the data was checked by the principal investigator on each day. After completing data collection, each questionnaire was checked for completeness and the correct entrance of the data was cheeked by the investigator.

## Data processing and analysis

Data cleaning and preparation was done to check for completeness of the data. The collected data were cleaned, coded, and entered into Epi info 7.1 and exported to the statistical package of social sciences (SPSS) version 20 for analysis. The descriptive statistics were summarized using summary statistics such as frequency tables, graphs, percentages, means, and standard deviations. To assess the influence of demographic, socio-demographic, environmental, and behavioral factors on pterygium, odds ratios (OR) were calculated with a significant test (p-value) and a 95% confidence level. Hosmer and Lemeshow goodness of fit was used to check for model fitness. Binary logistic regression was used to determine the association between the independent variable and pterygium. All variables in bivariable analysis with $p \leq 0.25$ were entered into a multivariable logistic regression to identify the factors associated with pterygium. Adjusted odds ratios along with 95% confidence interval in multivariable analysis, were used to determine the strength and direction of the statistical association between independent variables and pterygium.

## Ethical approval and consent to participate

Ethical clearance was obtained from the University of Gondar ethical review committee and a letter of permission was obtained from Gambella regional health office and Gambella town administrative office. Before starting the examination, written consent was taken from each study participant. They were given full right to participate and to refuse or withdraw at any

time they want. Confidentiality of the information obtained was assured by avoiding personal identifiers like a name from the data collection tool and also through coding and locking the data.

## Results

### Socio-demographic characteristics of the study participants

There were 400 study participants in the assessment of pterygium with a response rate of 99.50%. The mean age of the study participants was 39.9 ± 9.8years. More than half 219 (54.75%) of the study participants were males and slightly higher than two-third 273(68.25%) of the participants were living in rural areas.

Among the study participants, almost half 206(51.50%) were married and 179(44.80%) had a family size of 4–6 children in a house. Only 114 (28.50%) of the participants could read and write as well as 164(41.00%) were merchants (Table 1).

**Table 1. Sociodemographic characteristics of study participants in a study conducted to assess the prevalence of pterygium and its associated factors among adults aged 18 years and above in Gambella town, Southwest Ethiopia, 2019 (n = 400).**

| Variables | Frequency | Percent (%) |
|---|---|---|
| **Age group** | | |
| 20–40 years | 212 | 53.00 |
| 40–60 years | 176 | 44.00 |
| >60 years | 12 | 3.00 |
| **Sex** | | |
| Male | 219 | 54.80 |
| Female | 181 | 45.20 |
| **Marital status** | | |
| Married | 206 | 51.50 |
| Widowed | 99 | 24.80 |
| Divorced | 44 | 11.00 |
| Single | 51 | 12.70 |
| **Family size** | | |
| 0–3 | 141 | 35.20 |
| 4–6 | 179 | 44.80 |
| ≥ 7 | 80 | 20.00 |
| **Educational level** | | |
| Unable to read write | 56 | 14.00 |
| Able to read and write | 114 | 28.50 |
| Primary school | 113 | 28.30 |
| Secondary school | 94 | 23.40 |
| College/university | 23 | 5.80 |
| **Occupation** | | |
| Student | 24 | 6.00 |
| Farmer/labor | 40 | 10.00 |
| Employed | 153 | 38.30 |
| Merchant | 164 | 41.00 |
| Others * | 19 | 4.70 |

*Housewife, priests.

## Environmental and behavioral factors of study participants

More than half 212(53.00%) of the participants had a history of sun exposure for five and more hours per day. Whereas only 97(24.30%) use sunglass/hat for protection (Table 2). Besides, almost three-fourth of the study participants 294(73.40%) never use any alcohol product and only 63 (15.80%) had a family history of pterygium (Table 3).

## Prevalence of pterygium

The overall prevalence of pterygium among adults aged 18 years and above in Gambella town was 127(31.80%), (95% CI: 27.30, 36.30) in either eye (Fig 1). Most of the pterygium 101 (79.53%) was located to the nasal side of the cornea and the rest 26(20.47%) was located to the temporal part of the cornea. More than half 69(54.33%) of the pterygium was unilateral and the remaining 58(45.67%) was bilateral.

## Factors associated with pterygium

From the bi-variable logistic analysis 10 variables including; sex being male, address being rural, work area being outdoor, dust exposure, wearing of sunglass/hat, and occupation being

**Table 2. Environmental factors of study participants in a study conducted to assess the prevalence of pterygium and its associated factors among adults aged 18 years and above in Gambella town, Southwest Ethiopia, 2019 (n = 400).**

| Variables | Frequency | Percent |
|---|---|---|
| **Sun exposure** | | |
| Yes | 212 | 53.00 |
| No | 188 | 47.00 |
| **Working area** | | |
| Outdoor | 155 | 38.70 |
| Indoor | 245 | 61.30 |
| **Dust exposure** | | |
| Yes | 161 | 40.30 |
| No | 239 | 59.70 |
| **Sunglass use** | | |
| Yes | 97 | 24.30 |
| No | 303 | 75.70 |

**Table 3. Behavioral factors among study participants in a study conducted to assess the prevalence of pterygium and its associated factors among adults aged 18 years and above in Gambella town, Southwest Ethiopia, 2019 (n = 400).**

| Variables | Frequency | Percent |
|---|---|---|
| **Alcohol use** | | |
| Never | 294 | 73.40 |
| Past | 55 | 13.80 |
| Current | 51 | 12.80 |
| **Smoking** | | |
| Never | 319 | 79.70 |
| Past | 41 | 10.30 |
| Current | 40 | 10.00 |
| **History of traditional medication** | | |
| Yes | 58 | 14.50 |
| No | 342 | 85.50 |

# Percent

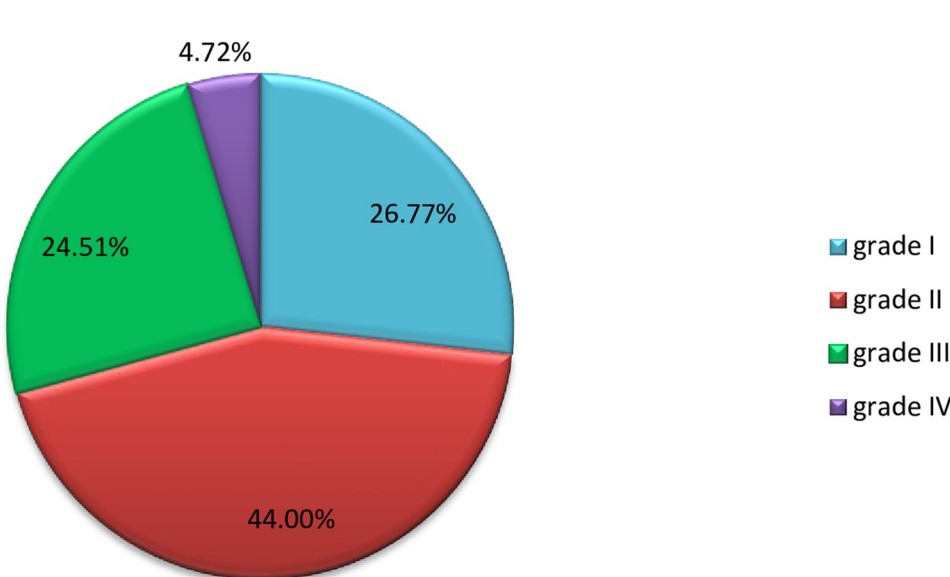

**Fig 1. Proportion and grades of pterygium among adults living in Gambella town, Southwest Ethiopia, 2019 (n = 127).**

farmer, low-level of education, smoking, sun exposure, and marital status being married were independently associated with the occurrence pterygium with a significant level of $p \leq 0.25$. The multivariable logistic analysis revealed that only four variables which include; sex being male, a working area being outdoor, sun exposure, and wearing of sunglass/hat were independently associated with the occurrence of pterygium.

The odds of developing pterygium among adult male participants were 2.10 times more likely than females (AOR = 2.10 (95% CI: 1.26, 3.45)). Those adults who were exposed to sunlight were 6.86 times more likely to develop pterygium as compared to those who were not exposed to sunlight (AOR = 6.86 (95% CI: 4.00, 11.79). Also, the odds of developing pterygium among adults with an outdoor working environment were 2.10 times more likely as compared to those with an indoor working environment (AOR = 2.10(95% CI: 1.21, 3.60)). On the other side, adults who were using sunglasses/hats for protection were 62% less likely to develop pterygium as compared to those who did not wear sunglasses/hats (Table 4).

## Discussion

The magnitude of pterygium varies widely from 1.1% to 53% globally [4–7]. This is the first community based cross-sectional study on pterygium in Gambella town among adults. This study indicated that the overall prevalence of pterygium among adults in either eye was 127 (31.80%), (95% CI: 27.3, 36.3).

This finding is higher as compared to other community based studies done in a China (17.9%), (6.4%), and (11.75%) [4, 30, 31], Indonesia (10.00%) [32], Iran (9.4%), (1.3%), (11%), (13.11%), and (14.99%) [3, 14, 33, 34], Central Myanmar (19.6%) [13], South Korea (8.8%) and (6.7%) [10, 24], Malaya Singapore(12.3%) [35], India (13%) and (11.7%) [20, 36], Northern Japan (4.4%) [37], Spain (5.9%) [38] and Kolla diba, Ethiopia (18.5%) [23]. The possible reason behind our high finding could be it is carried out in a hot, dry, and dusty environment with low latitude that might contribute to the occurrence of pterygium [39, 40]. Furthermore,

**Table 4. Factors associated with pterygium in a study conducted to assess the prevalence of pterygium and its associated factors among adults aged 18 years and above in Gambella town, Southwest Ethiopia, 2019 (n = 400).**

| Variables | Pterygium | | COR (95% CI) | AOR (95% CI) |
|---|---|---|---|---|
| | Yes | No | | |
| **Sex** | | | | |
| Female | 54 | 165 | 1.00 | |
| Male | 73 | 108 | 2.07(1.35,3.17) | **2.10(1.26, 3.45)** ** |
| **Address** | | | | |
| Urban | 49 | 79 | 1.00 | |
| Rural | 78 | 195 | 1.57(1.01,2.45) | |
| **Dust exposure** | | | | |
| No | 81 | 154 | 1.00 | |
| Yes | 46 | 119 | 1.36 (0.88, 2.10) | |
| **Sun exposure** | | | | |
| No | 94 | 94 | 1.00 | |
| Yes | 33 | 179 | 5.42(3.40, 8.67) | **6.86(4.00, 11.79)** *** |
| Marital status | | | | |
| Single | 18 | 33 | 1.00 | |
| Married | 53 | 153 | 1.58(0.82, 3.03) | |
| Widowed | 36 | 63 | 0.96(0.47, 1.93) | |
| Divorced | 20 | 24 | 0.66(0.29, 1.50 | |
| **Occupation** | | | | |
| Student | 9 | 15 | 1.00 | |
| Farmer | 9 | 31 | 2.10(0.68, 6.28) | |
| Employed | 46 | 107 | 1.40(0.57, 3.42) | |
| Merchant | 55 | 109 | 1.19(0.49, 2.89) | |
| Others* | 8 | 11 | 0.83(0.24, 2.82) | |
| **Working area** | | | | |
| Indoor | 90 | 155 | 1.00 | |
| Outdoor | 37 | 118 | 1.85(1.18, 2.91) | **2.10(1.21, 3.60)** ** |
| **Smoking** | | | | |
| Never | 96 | 223 | 1.00 | |
| Past | 19 | 22 | 0.50(0.26, 0.98) | |
| Current | 12 | 28 | 1.00(0.49, 2.05) | |
| **Educational level of house head** | | | | |
| Unable to write &read | 13 | 43 | 2.54 (0.91, 7.12) | |
| Primary school | 61 | 166 | 2.10 (0.87, 5.00) | |
| Secondary school | 43 | 51 | 0.91(0.36, 2.29) | |
| College/university | 10 | 13 | 1.00 | |
| **Educational level recoded** | | | | |
| Lower level of education | 114 | 230 | 1.64(0.85, 3.17) | |
| Higher level of education | 13 | 43 | 1.00 | |
| **Sunglass/hat use** | | | | |
| Yes | 107 | 192 | 0.44(0.26, 0.76) | **0.39(0.21, 0.73)** ** |
| No | 20 | 81 | 1.00 | |

**Abbreviations:** COR = Crude odds ratio, AOR = Adjusted odds ratio, *House wife, priests ** P-value <0.05, *** p<0.001.

different geographical areas nearer to the equator (tropical area) have greater exposure to ultraviolet-B that intern contributes to the occurrence of pterygium [12, 19].

The proportion of pterygium in this study is similar to other studies done in Southwest, Japan (30.8%) [12], China (33.98%) [39]. Although these countries are located in different geographical areas, the possible reason might be all studies were carried out with similar study design.

On the other side, the proportion of pterygium in this study is lower than other studies done in Northwest Ethiopia (38.7%) [8]. This might due to geographical, latitudinal, economic, and environmental variations that contribute to the occurrence of the disease [19, 23]. The other possible reason could due to the age groups included in the studies. Adults aged 18 years and above were included in this study but, the study done in Gondar, Ethiopia includes adults age greater than 20 years that might contribute to the occurrence of the pterygium [4, 19, 21].

This study also indicated that the proportion of pterygium by its clinical appearance as grade I to IV. As a result, grade II pterygium was most common 56(44.00%) followed by grade I 34(26.70%). In addition, most of the pterygium 101(79.53%) was located on the nasal side of the cornea. In other works of literature, the most common reported grades were grade II both in Mongolian, China [4] (162/2651, 6.1%) and in Xinjiang, China [30] 258 (5.59%), grade I both in Gondar, Ethiopia [8] 97(24.87%) and Koladiba Ethiopia [23] 63(56.3%). In addition, most of the pterygium was located to the nasal side of the cornea in Mongolian, China [4] (38/2651, 1.4%), Bhagalpur, India [11] (93.49%), Gondar, Ethiopia [8] 149(98.7%), Koladiba Ethiopia [23] 87 (77.7%) and Xinjiang, China [30] (11.54%). Even though there is no clear justification for similarities and discrepancies of these results, different environmental areas, age groups included, study design applied, climatic conditions and UV exposure status of study participants might contribute to the disease conditions [6, 19, 21, 40].

The present study revealed that odds of developing pterygium among those working in an outdoor environment were 2.10 times more likely as compared to an indoor environment (AOR = 2.10 (95% CI: 1.21, 3.60)). This is supported by many studies done in Ethiopia [8, 23], Myanmar [13], Indonesia [25, 35], Japan [12], Iran [14], Central India [20], Mongolia China [4]. This might due to outdoor workers had a prolonged period of UV light as well as dust exposure as compared to indoor workers [4, 13, 19, 23].

Those adults who were exposed to sunlight five hours and above were 6.86 times more likely to develop pterygium than those who were not exposed (AOR = 6.86 (95% CI: 4.00, 11.79). This is in line with studies done in South Korea [24], Australia [5], India [36], Koladiba Ethiopia [23]. Although it is difficult to quantify the true amount of one's exposure to sunlight with ultraviolet radiation, many studies had shown that outdoor workers have a higher risk of the development of pterygium [2, 24]. Even though some amount of ultraviolent is important for our body, exposure for five hours and more for acute or long time could cause disorders in skin, conjunctiva, cornea, and lens [2, 9, 19].

Lots of controversies are raised regarding the association of gender and pterygium in different kinds of literature. This study showed that being male was 2.10 times more likely to develop pterygium as compared to females. This result is supported by studies done in Ethiopia [8], Australia [5], China [1, 26, 41], Japan [12], Malaya Indonesia [35], and Central India [20]. Males spent most of their time in outdoor activities in most African countries including Ethiopia. This might intern expose them to develop pterygium [8]. Whereas a study in South Korea [24] showed that being female was associated with pterygium. In contrast, studies in Japan [37], China [4, 30], and Iran [34] indicated that there were no significant differences in the association of gender and pterygium. Cultural differences, differences in ethnicity, and

varied environmental factors could affect the exposure status of males and females for pterygium [21].

On the other side, adults who wore sunglass/hat were 62% less likely to develop pterygium as compared to those who did not wear sunglass/hat. This is consistent with studies done in Gondar, Ethiopia [8], and Benin City, Nigeria [42]. The possible reason could be wearing hat/ protective sunglasses might reduce the exposure status from UV light [36] and dust particles that might also reduce the development of pterygium [5].

## Conclusion

The prevalence of pterygium was high among adults living in Gambella town. Genders being male, outdoor working area, and sunlight exposure were positively associated with the development of pterygium. Whereas wearing sunglass/hat was a protective factor for the development of pterygium. Sunlight exposure and outdoor activities were the only modifiable risk factors.

## Supporting information

**S1 Table. Multivariable analysis final out put in a study to assess prevalence of pterygium and its associated factors among adults aged 18 years and above in Gambella town, Southwest Ethiopia.**
(DOCX)

**S1 File. Questionnaire and data extraction MS format to the study of prevalence of pterygium and its associated factors among adults aged 18 years and above in Gambella town, Southwest Ethiopia.**
(DOCX)

**S2 File. Sav original data showing the association of different variables in study to assess the prevalence of pterygium and its associated factors among adults aged 18 years and above in Gambella town, Southwest Ethiopia.**
(SAV)

## Acknowledgments

We would like to acknowledge the Gambella regional health office and Gambella town administrative office for their technical support to carry out this research.

## Author Contributions

**Conceptualization:** Sofonias Addis Fekadu.

**Data curation:** Abel Sinshaw Assem.

**Formal analysis:** Sofonias Addis Fekadu, Nebiyat Feleke Adimassu.

**Investigation:** Sofonias Addis Fekadu.

**Methodology:** Abel Sinshaw Assem.

**Software:** Sofonias Addis Fekadu, Nebiyat Feleke Adimassu.

**Supervision:** Sofonias Addis Fekadu, Abel Sinshaw Assem.

**Validation:** Sofonias Addis Fekadu, Abel Sinshaw Assem, Nebiyat Feleke Adimassu.

**Visualization:** Sofonias Addis Fekadu, Nebiyat Feleke Adimassu.

**Writing – original draft:** Sofonias Addis Fekadu.

**Writing – review & editing:** Abel Sinshaw Assem, Nebiyat Feleke Adimassu.

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
