## [Decision Letter · Decision Letter 0]

23 Jul 2020

PONE-D-20-18073

Prevalence of pterygium and its associated factors among adults aged 18 years and above in Gambella town, Southwest Ethiopia, May 2019

PLOS ONE

Dear Dr. Fekadu,

Thank you for submitting your manuscript to PLOS ONE. The reviewers are very complimentary of you manuscript, but there are a few minor clarifications to address

Therefore, we invite you to submit a revised version of the manuscript that addresses the points raised during the review process.

We look forward to receiving your revised manuscript.

Kind regards,

James Wolffsohn, PhD

Academic Editor

PLOS ONE

Journal Requirements:

https://journals.plos.org/plosone/article?id=10.1371%2Fjournal.pone.0215528

In your revision ensure you cite all your sources (including your own works), and quote or rephrase any duplicated text outside the methods section. Further consideration is dependent on these concerns being addressed.

"The funders had no role in study design, data collection and analysis, decision to publish, or preparation of the manuscript"

6. Your ethics statement must appear in the Methods section of your manuscript. If your ethics statement is written in any section besides the Methods, please move it to the Methods section and delete it from any other section. Please also ensure that your ethics statement is included in your manuscript, as the ethics section of your online submission will not be published alongside your manuscript.

Reviewers' comments:

Reviewer's Responses to Questions

**Comments to the Author**

1. Is the manuscript technically sound, and do the data support the conclusions?

Reviewer #1: Yes

Reviewer #2: Yes

2. Has the statistical analysis been performed appropriately and rigorously? 

Reviewer #1: Yes

Reviewer #2: Yes

3. Have the authors made all data underlying the findings in their manuscript fully available?

Reviewer #1: Yes

Reviewer #2: Yes

4. Is the manuscript presented in an intelligible fashion and written in standard English?

Reviewer #1: Yes

Reviewer #2: Yes

5. Review Comments to the Author

Reviewer #1: As the authors note, there are a number of peer-reviewed publications on the epidemiology of pterygium. Sun exposure is causal. The data in this paper further confirm previous findings of risk factors in a unique population-based study. A few suggestions/questions:

1. How was dust exposure defined?

2. How was sunglass use defined? Did participants have to wear them for a certain percent of tie while outdoors to be considered 'yes'?

3. Self-reported family history of pterygium is not likely to be accurate and not relevant to this paper, so I suggest deleting.

4. p<0.25 is an unusual significance level to use. How do the results change is a p<0.10 is used?

5. The independent variables in the multivariate analysis are likely highly corrected, this making interpretation challenging. How was that taken into account?

6. If the data are available, it would be interesting to assess a possible dose response relationship between sun exposure and size of pterygium.

Reviewer #2: I found this article well written and the topic interesting. I applaud the authors for the diligence and success in gathering data for this study. Though not material to the work reported here, I think the author would also be interest in exploring the work on peripheral light focusing by Kwok and Coroneo for future research, as it would provide illumination regarding the impact of wearing sunglasses on exposure related to pterygium

6. PLOS authors have the option to publish the peer review history of their article (what does this mean?). If published, this will include your full peer review and any attached files.

Reviewer #1: **Yes: **Catherine McCarty Wallerstein

Reviewer #2: **Yes: **Cristina Schnider

---

## [Author Response · Author response to Decision Letter 0]

1 Aug 2020

Response to reviewers

Manuscript title: Prevalence of pterygium and its associated factors among adults aged 18 years and above in Gambella town, Southwest Ethiopia, May 2019

Manuscript number: PONE-D-20-18073

Dear reviewers. Thank you for giving us the chance to revise and correct the manuscript accordingly. We are thankful so much and we appreciate you for careful and in-depth reading of this manuscript and for the thoughtful comments and constructive suggestions, which help us to improve the quality of this manuscript. Saying this we addressed all the concerns raised by you and incorporated our reflection in the revised manuscript. We tried to address all issues that are raised to be addressed.

 Many thanks. 

Reviewer reports:

Reviewer #1(Catherine McCarty Wallerstein): As the authors note, there are a number of peer-reviewed publications on the epidemiology of pterygium. Sun exposure is causal. The data in this paper further confirm previous findings of risk factors in a unique population-based study. A few suggestions/questions: 1. How was dust exposure defined?

Authors’ Response: Thank you! We accepted the comment and we included in the “operational definition” parts of the manuscript. Dust exposure: a person having at least a history of one episode of exposure to dusty environment that causes their eye to be irritated, itchy and red within the past 6 months(1). 

Reviewer #1(Catherine McCarty Wallerstein): 2. How was sunglass use defined? Did participants have to wear them for a certain percent of tie while outdoors to be considered 'yes'?

Authors’ Response: Thank you! We accepted the comment and included in the operational definition part of the new revised manuscript. Sun glass use was defined as when participants used any type of glass(cosmetics, general and specially treated) for the purpose of protection from sun during their outdoor activities(2,3). 

Reviewer #1(Catherine McCarty Wallerstein): 3. Self-reported family history of pterygium is not likely to be accurate and not relevant to this paper, so I suggest deleting.

Authors’ Response: Thank you! for the comment, the comment is accepted and we deleted it from the new edited manuscript.

Reviewer #1(Catherine McCarty Wallerstein): 4. p<0.25 is an unusual significance level to use. How do the results change is a p<0.10 is used?

Authors’ Response: Thank you! For the comment.

 commonly used cutting point in bi-variable analysis to multivariable logistic regression is P< 0.20, but some statistician advice that when the variables that enter to multivariable analysis are fewer, it may possible to use the cutting point p<0. 25 and if we tried to use p< 0.10, variables will further be reduced and few not more than five will full fill the inclusion criteria which might reduce the precision of the study rather it is possible to enter all variables. in addition, the criteria developed in cutting the P-value in bivariable analysis is simply to reduce the number of variables that will inter to multivariable analysis. So, it is difficult to use p<.10, in our study since it will affect the whole outcome as well as statistical concept of the study. 

Reviewer #1(Catherine McCarty Wallerstein): 5. The independent variables in the multivariate analysis are likely highly corrected, this making interpretation challenging. How was that taken into account.

Authors’ Response: Thank you! We write in such a way that, in order to be easily understand by the readers and the nature of the variables can not be compressed to a single word easily and we tried to write the manuscript so that the words can easily be understandable. 

Reviewer #1(Catherine McCarty Wallerstein): 6. If the data are available, it would be interesting to assess a possible dose response relationship between sun exposure and size of pterygium. 

Authors’ Response: Thank you! Since sun exposure was operationalized as “those who had exposure for five hours and above are considered as exposed and others as non-exposed” no separate dose of hour was included in the data. So, it is difficult to show the relationship between dose of sun exposure with size of pterygium.

Reviewer #2(Cristina Schnider): Thank you for your constructive comment and suggestion, we will try to conduct a future research in the topic/idea you suggested us and we hope we may face you again with that title. 

Responses to accadamic editors 

1. Response of authors: thank you, we tried to correct the manuscript to meet the Plos one’s criteria of manuscript formatting as much as possible.

2. Response of authors: thank you for your comment, we have already rephrased and correct the duplicated and overlapping texts in the new edited manuscript that were closer to the previous publication.

3. Response of authors: thank you,

A. As already mentioned previously, there were no a piece of financial and material support for this study, even the organization that employed us. 

B. The funders had no role in the study design, data collection and analysis, decision to publish, or preparation of the manuscript. 

C. None of the authors received a salary from funders.

D. The authors received no specific funding for this work.

4. Thank you!

A. There is no ethical or legal restriction on sharing the data requested from us

B. Data are uploaded as “supporting information files” at the end of the new edited manuscript.

5. My ORCID iD is already updated from my PLOSE ONE account.

6. The ethics statement is already moved to method part in the new edited manuscript 

References for comments reviewer # 1 

1. De Smedt SK, Nkurikiye J, Fonteyne YS, Tuft SJ, Gilbert CE, Kestelyn P. Vernal keratoconjunctivitis in school children in rwanda: Clinical presentation, impact on school attendance, and access to medical care. Ophthalmology [Internet]. 2012;119(9):1766–72. Available from: http://dx.doi.org/10.1016/j.ophtha.2012.03.041

2. Backes C, Religi A, Moccozet L, Behar-Cohen F, Vuilleumier L, Bulliard JL, et al. Sun exposure to the eyes: predicted UV protection effectiveness of various sunglasses. J Expo Sci Environ Epidemiol [Internet]. 2019;29(6):753–64. Available from: http://dx.doi.org/10.1038/s41370-018-0087-0

3. Chorley AC, Evans BJW, Benwell MJ. Solar eye protection practices of civilian aircrew. Aerosp Med Hum Perform. 2015;86(11):953–61.

---

## [Editor Report · Decision Letter 1]

5 Aug 2020

Prevalence of pterygium and its associated factors among adults aged 18 years and above in Gambella town, Southwest Ethiopia, May 2019

PONE-D-20-18073R1

Dear Dr. Fekadu,

We’re pleased to inform you that your manuscript has been judged scientifically suitable for publication and will be formally accepted for publication once it meets all outstanding technical requirements.

Kind regards,

James Wolffsohn, PhD

Academic Editor

PLOS ONE

Additional Editor Comments (optional):

Thank you for carefully incorporating the reviewers comments to enhance your manuscript.
---

## [Editor Report · Acceptance letter]

20 Aug 2020

PONE-D-20-18073R1 

Prevalence of pterygium and its associated factors among adults aged 18 years and above in Gambella town, Southwest Ethiopia, May 2019 

Dear Dr. Fekadu:

I'm pleased to inform you that your manuscript has been deemed suitable for publication in PLOS ONE. Congratulations! Your manuscript is now with our production department. 

Kind regards, 

on behalf of

Professor James Wolffsohn 

Academic Editor

PLOS ONE